# Strain-controlled shell morphology on quantum rods

Botao Ji [1,2], Yossef E. Panfil[1,2], Nir Waiskopf[1,2], Sergei Remennik[2], Inna Popov[2] & Uri Banin [1,2]

Semiconductor heterostructure nanocrystals, especially with core/shell architectures, are important for numerous applications. Here we show that by decreasing the shell growth rate the morphology of ZnS shells on ZnSe quantum rods can be tuned from flat to islands-like, which decreases the interfacial strain energy. Further reduced growth speed, approaching the thermodynamic limit, leads to coherent shell growth forming unique helical-shell morphology. This reveals a template-free mechanism for induced chirality at the nanoscale. The helical morphology minimizes the sum of the strain and surface energy and maintains band gap emission due to its coherent core/shell interface without traps, unlike the other morphologies. Reaching the thermodynamic controlled growth regime for colloidal semiconductor core/shell nanocrystals thus offers morphologies with clear impact on their applicative potential.

---

[1] Institute of Chemistry, The Hebrew University of Jerusalem, Jerusalem 91904, Israel. [2] The Center for Nanoscience and Nanotechnology, The Hebrew University of Jerusalem, Jerusalem 91904, Israel. Correspondence and requests for materials should be addressed to U.B. (email: uri.banin@mail.huji.ac.il)

Colloidal semiconductor quantum heterostructures combine epitaxially grown interfaces between two semiconductor materials and are the basis of numerous optoelectronic devices. Combinations of semiconductors with a suitable energy band alignment may be chosen for a desired functionality, such as in a display, as laser materials, for photodetection or as photocatalysts[1–5]. Core/shell architectures are central to the successful realization of such applications[6]. To avoid interfacial defects, growth conditions close to the thermodynamic limit are preferred. Typically, flat shell growth is desired for surface passivation, in analogy to quantum wells grown via precursor deposition from the gas phase as applied in Molecular Beam Epitaxy (MBE) layer-by-layer for two semiconductors with small lattice mismatch[7]. Large lattice mismatch leads to another form of thermodynamic growth, yielding self-assembled quantum dots (QDs) in MBE through the Stranski-Krastanov (SK) mechanism in which at first, a continuous film of the second semiconductor (up to several monolayers) can be deposited (wetting layer)[7,8]. The strain energy induced by the lattice mismatch between the two materials accumulates and above a critical thickness, three-dimensional island growth will be favored to relieve the misfit strain, leading to QDs formation.

SK growth on colloidal quantum heterostructures is intriguing due to the reduced dimensionality and the nanoscale dimensions of the surfaces and so far, SK growth on colloidal QDs or quantum wires led to disordered islands limiting their functionality[9–12]. Herein, we report well-controlled SK type shell growth in core/shell quantum rods. By controlling the shell growth rate, transformation between a flat shell, to a shell with close lying islands, and remarkably to a shell with a helical ordered arrangement of the shell islands is demonstrated (Fig. 1a). Beyond the fundamental discovery of the ordered islands shell growth, such structures for ZnSe/ZnS quantum rods provide stable functionality for the heterostructures as heavy-metal-free photoinitiators and yield enhanced photoluminescence properties.

## Results

### Controlling shell morphologies of ZnS on ZnSe nanorods.

ZnSe nanorods passivated by organic ligands were synthesized firstly via oriented attachment, to serve as a model system[13]. The starting ZnSe nanorods are single-crystalline with a typical hexagonal wurtzite structure, ~30 nm in length and ~4.0 nm in diameter (Supplementary Fig. 1). The elongated dimension of quantum rods typically leads to a significant fraction of unpassivated surface-related defects[14,15]. Upon excitation, the generated excitons can diffuse along the long axis and one or both of the charge carriers (electron, hole) may be trapped by the surface defects, consequently degrading the properties of quantum rods and emphasizing the essential development of suitable shell growth modalities. The growth of ZnS shell on ZnSe nanorods was performed using shell precursors with low reactivity (alkylthiol and metal oleate) which were slowly introduced at elevated temperature[16,17]. The thiol group can bind strongly to soft metal ions on the surface of nanorods. The metal oleate induces the cleavage of the strong carbon-sulfur bond in the alkylthiol at high temperature, resulting in uniform shell growth on these anisotropic nanocrystals.

Purified ZnSe nanorods were redispersed in a mixture of 1-octadecene, oleylamine and oleic acid. The ZnS shell was grown by slowly injecting zinc oleate and 1-octanethiol separately at 310 °C. The reactivity of zinc oleate was adjusted in different experiments by changing the molar ratio between zinc and oleic acid as a powerful means to manipulate the shell growth morphology[18]. Larger excess of oleic acid can increase the solubility of the zinc precursor and thus limit zinc availability, producing zinc precursor with lower reactivity. When zinc oleate with a 1/4 molar ratio between zinc and oleic acid was used, the thickness of the ZnS shell steadily increased upon the addition of precursors (Supplementary Fig. 2). Free ZnS nanoparticles from self-nucleation indicate high reactivity of zinc oleate. A substantially flat ZnS shell morphology is observed by transmission electron microscopy (TEM) (Fig. 1b, termed as core/flat-shell nanorods for simplification). Hypothetically, this is a result of the kinetically controlled shell growth process in these conditions, which will be further justified.

By reducing the reactivity (using zinc oleate with a 1/6.3 molar ratio between zinc and oleic acid)[18], ZnSe nanorods with a thin ZnS shell are obtained initially in the first 60 min (Fig. 1c). The diameter of the nanorods increased by 1.1 nm to ~5.1 nm (Supplementary Fig. 3a and 4b). High-resolution TEM (HRTEM) image shows that the shell grew in an epitaxial way. The ZnS shell maintained uniformity upon further addition of shell precursors (reaction time, 90 min). The diameter of the nanorods increased to 6.2 nm, corresponding to a shell thickness of ~3.3 monolayers of ZnS (the thickness of one monolayer of wurtzite ZnS along the [100] growth direction is 0.33 nm) (Supplementary Fig. 3b, 4c and 5). Further ZnS shell growth produced nanorods with the appearance of small bulges on the surface, indicating the inhomogeneity of the shell thickness (Supplementary Fig. 3c, reaction time, 120 min). Adding more shell precursors led to thicker nanorods with increased roughness of the nanorod surface (Supplementary Fig. 3d and e, reaction times between 150 and 180 min). When the amount added was equivalent to the flat shell case (Fig. 1b) discussed above, a zigzag ZnS shell structure was obtained (Fig. 1d and Supplementary Fig. 3f, reaction time, 210 min).

The high angle annular dark field (HAADF) scanning TEM (STEM) image shows the architecture of the obtained nanorods, where three-dimensional islands are clearly visible (Fig. 1e, f). Electron scattering intensity in HAADF Z-contrast imaging is approximately proportional to $Z^{1.6–1.9}$ where Z is the atomic number[19]. Thereby, the Se atoms from ZnSe core can be easily distinguished from the ZnS shell due to the relatively low scattering intensity of the lightest S atoms. A central region of high contrast is observed in all nanorods, attributed to the straight ZnSe core rods, consistent with the higher Z of selenium versus sulfur. EDX elemental mapping in STEM in the same area further confirms that Se atoms are distributed in the inner core part, while S atoms can be identified in the outer shell. Zn atoms are distributed homogeneously throughout the nanorods (Fig. 1f–i). Powder X-ray diffraction (XRD) measurements indicate that the original wurtzite crystal structure of ZnSe nanorods is maintained, while all the peaks are shifted to higher angles upon the ZnS shell growth, consistent with its smaller lattice constant compared with ZnSe (Supplementary Fig. 6).

Further slowing down the growth rate will allow the shell growth to approach the thermodynamic limit. This was achieved via using even less reactive precursor prepared by adding larger excess of oleic acid (molar ratio, Zn/oleic acid, 1/10). Initially, a uniform ZnS coverage was observed as the wetting layer grown on the ZnSe nanorods (Supplementary Fig. 7a). Addition of more shell precursors led to the appearance of bulges (Supplementary Fig. 7b and c). Notably, the distance between the bulges is larger than that in the case of the islands-shell growth. Further growth of ZnS shell induced the development of a thread with large pitch on the surface, producing ZnSe/ZnS nanorods with unique helical-shell morphology (Fig. 1j and Supplementary Fig. 8). The helical shell became more emphasized upon further growth of ZnS (Supplementary Fig. 7f). Corresponding XRD pattern matches the hexagonal wurtzite structure (Supplementary Fig. 9). The HAADF STEM images clearly show the thread of the ZnS shell with lighter contrast (Fig. 1k, l). EDX elemental mapping in STEM confirms that Se and S atoms are located in the inner core part and the outer helical shell, respectively (Fig. 1l–o). While helical motifs in

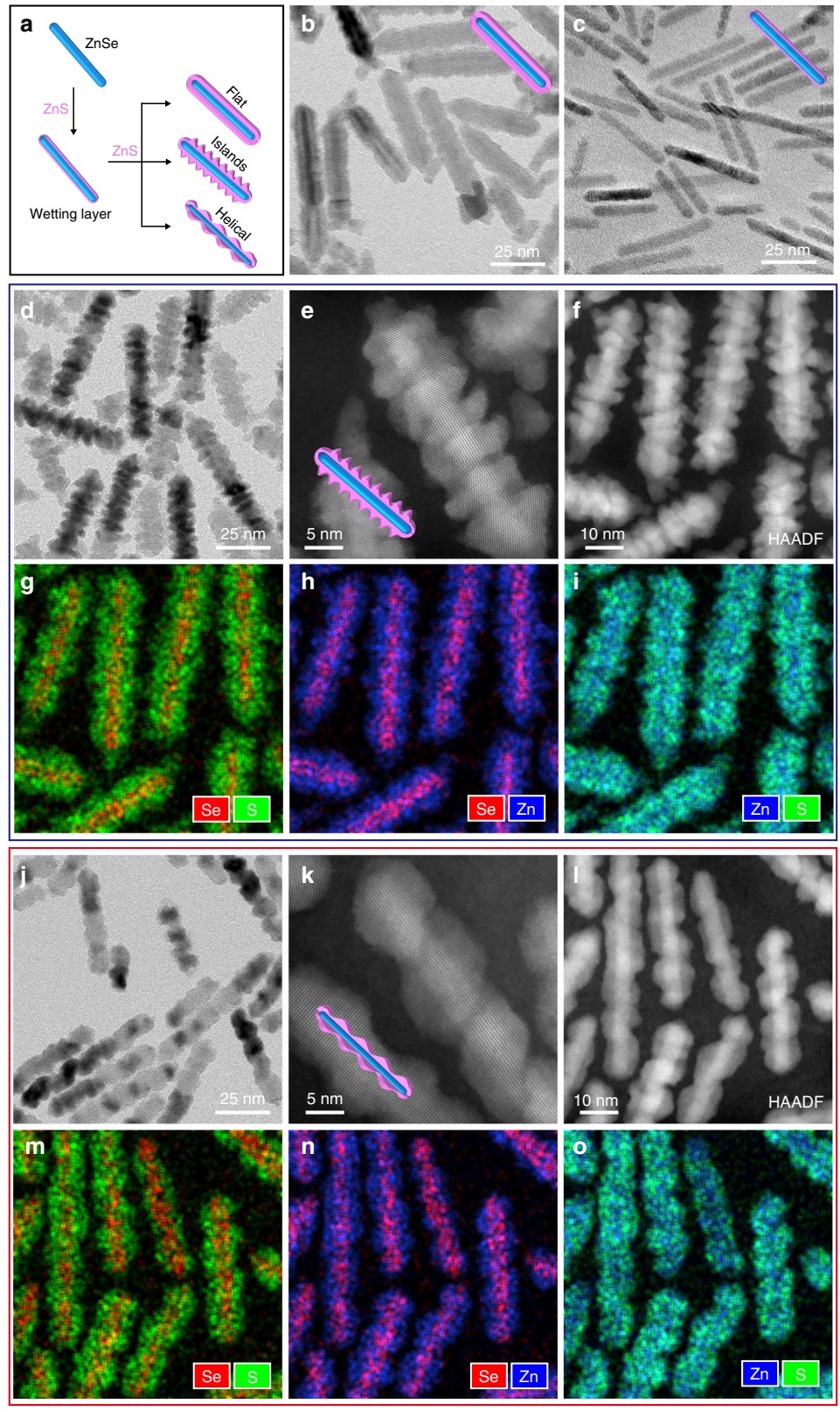

semiconductor nanorods and nanowires were reported through oriented attachment-, light-, chiral ligands- and magnetic field-induced synthesis[20–24], to the best of our knowledge, our study presents the first example of helical-shell growth on semiconductor nanorods.

**Generality of islands-shell growth on nanorods**. We next focus our attention on the intermediate case of islands shell growth mode. Atomic resolution HAADF-STEM imaging enables insight on the structure of ZnSe/ZnS core/islands-shell nanorods (Fig. 2a). Inspection of the magnified image of area b in Fig. 2a

**Fig. 1** Controlling shell morphologies of ZnS on ZnSe nanorods. **a** Schematic illustration of the controlled shell growth of ZnS on a ZnSe nanorod: Step I, initial uniform ZnS shell growth when below critical thickness. Step II, further ZnS growth leads to flat-, islands- and helical-shell morphologies by controlling the ZnS growth rate via tuning the precursor reactivity. **b** TEM image of ZnSe/ZnS core/flat-shell nanorods in step II. **c** TEM image of ZnSe nanorods with thin uniform ZnS shell before islands growth starts. **d**, **e** TEM and high-resolution high angle annular dark field (HAADF) scanning TEM (STEM) images of ZnSe/ZnS core/islands-shell nanorods in step II. **f-i** HAADF STEM image of ZnSe/ZnS core/islands-shell nanorods and corresponding overlay mapping of two elements of multiple nanorods based on Energy dispersive X-ray spectroscopy (EDX) scan. **g** Se and S. **h** Se and Zn. **i** Zn and S. **j**, **k** TEM and high-resolution HAADF STEM images of ZnSe/ZnS core/helical-shell nanorods in step II. **l-o** HAADF STEM image of ZnSe/ZnS core/helical-shell nanorods and corresponding overlay mapping of two elements of multiple nanorods. **m** Se and S. **n** Se and Zn. **o** Zn and S. The samples shown in **b**, **d** and **j** were obtained by adding similar amounts of shell precursors with different reactivity

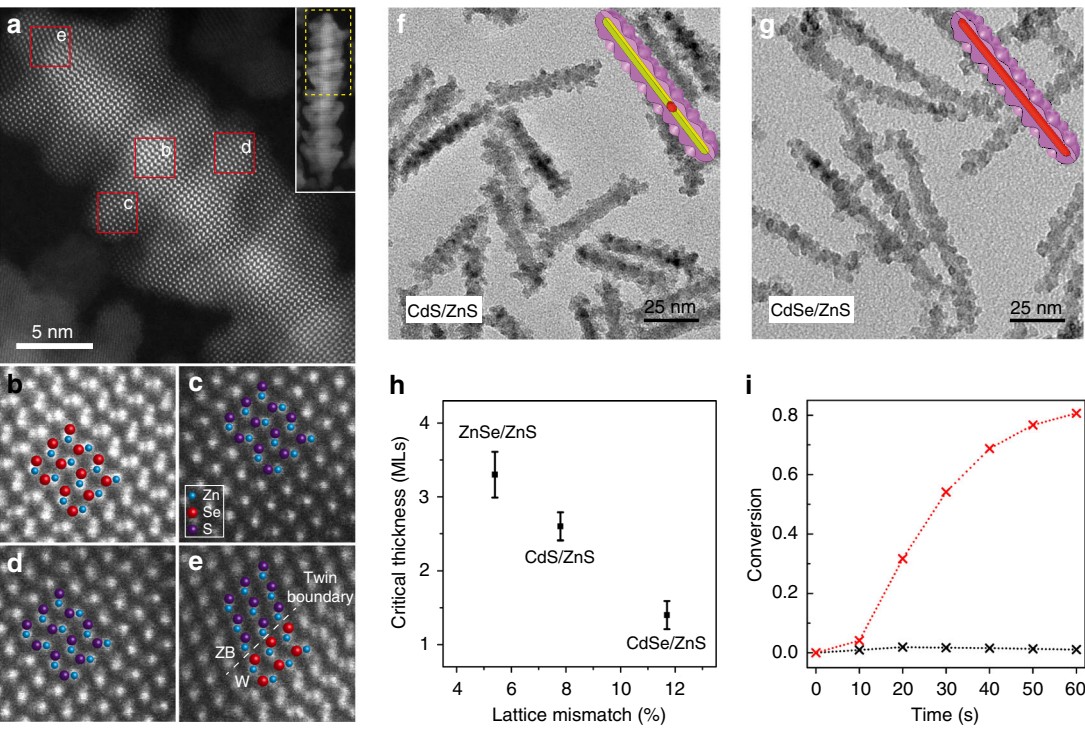

**Fig. 2** The generality of three-dimensional islands-shell growth of ZnS on different nanorods. **a** High-resolution HAADF STEM images of ZnSe/ZnS core/islands-shell nanorods. Inset shows the whole nanorods, while the yellow dash rectangle is magnified in **a**. **b-e** Zoom in images of areas indicated in **a**. Zn atoms are marked in blue, Se atoms in red and S atoms in purple. **f**, **g** TEM images of CdSe/CdS/ZnS and CdSe/ZnS core/islands-shell nanorods. **h** The dependence of the critical thickness of the wetting layer on lattice mismatch, showing its inverse relation. Error bars (s.d.) originate from the distribution of critical thickness. **i** Polymerization degree of acrylamide under UV light at 405 nm with intensity of 30 mW/cm$^2$ using ZnSe (black) and ZnSe/ZnS core/islands-shell nanorods (red) as photoinitiators in aerobic conditions. The ZnS shell growth is needed to stabilize the rods allowing for its functionality

indicates that the original hexagonal structure of ZnSe is not altered after ZnS shell growth. Equivalent analyses focusing on the shell at different regions (areas c and d in Fig. 2a) reveal the epitaxial ZnS shell growth showing atomic arrays coherent with the ZnSe core. Notably, cubic ZnS shell growth can be recognized on the apexes of the ZnSe nanorod, creating a twin boundary along the (002) plane (Fig. 2e). While considering the location of the islands, we performed HRTEM analyses which excluded the possibility that the formation of islands-shell is related to the stacking faults in the core rod (Supplementary Fig. 10 and 11). It should be noted that Oh et al.[12] reported on islands-decorated nanostructure by growing CdSe QDs on CdS nanorods anisotropically etched by zinc oleate. However, in the present work the shell precursors were added slowly at high temperature and then reacted for the shell growth, thus the actual concentrations of both shell precursors in the solution were low. Additionally, the ZnSe nanorods studied here possess the same cation ions as the zinc precursor and therefore the addition of zinc oleate will suppress their etching. Therefore etching is not a significant process during the synthesis of ZnSe/ZnS core/islands-shell.

The islands-shell growth of ZnS resembles the characteristics of the SK growth but on the finite rod surface. In this process, slowing the ZnS growth rate plays a key role in allowing dislocation nucleation of islands to release the interfacial strain energy unlike fast growth that results in the flat shell morphology (Fig. 1b). The mechanism of this type of growth reactivity control on shell growth is further substantiated by studying CdS shell growth on CdSe, ZnS, and ZnSe nanorods using cadmium oleate with a 1/6.3 molar ratio between cadmium and oleic acid (the same molar ratio as in the case of ZnS islands-shell growth). Relatively flat CdS shells were obtained in all cases (Supplementary Fig. 12) even when the CdS shell was very thick. The higher reactivity of cadmium oleate than zinc oleate[12] leads to the fast deposition of CdS shell, overcoming the dislocation nucleation by the formation of islands, indicating the flat-shell formation is a result of kinetically controlled shell growth.

To expand the family of such unique core/islands-shell architectures and aiming to demonstrate the generality of this growth behavior in colloidal semiconductor nanocrystals, we

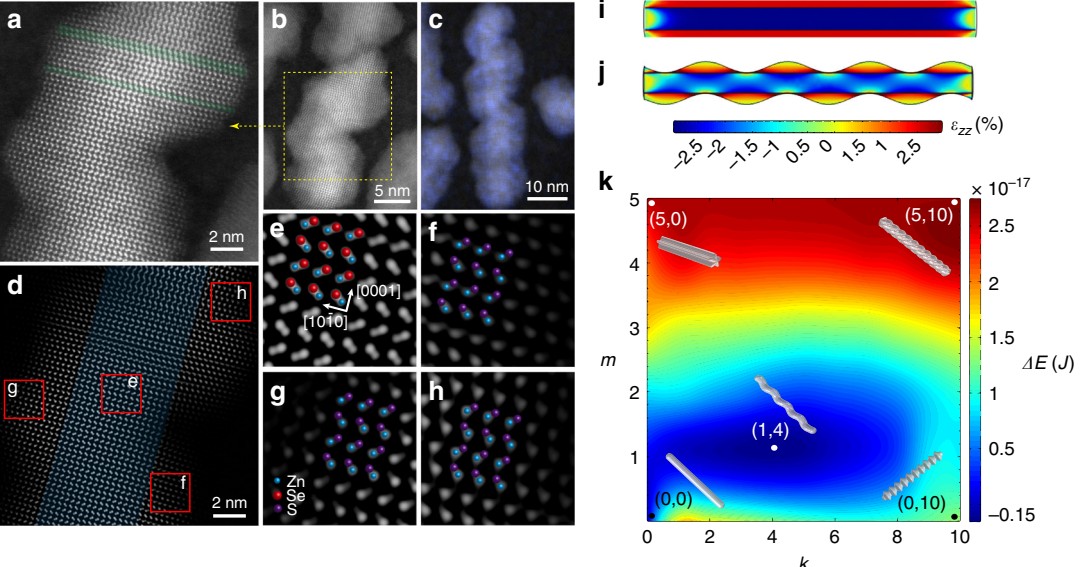

**Fig. 3** Helical-shell growth of ZnS on ZnSe nanorods. **a**, **b** High-resolution HAADF STEM images of ZnSe/ZnS core/helical-shell nanorods. Stacking faults are marked in green in (**a**) and are not correlated with the helical pitch. **c** HAADF STEM image of ZnSe/ZnS core/helical-shell nanorods overlaid with Zn elemental mapping. **d** Inverse fast Fourier transform (IFFT) filtered image corresponding to **a**. **e**–**h** Magnified images of the areas indicated in **d**. Zn atoms are marked in blue, Se atoms in red and S atoms in purple. Coherent growth of the lattice is identified. **i** The calculated elastic strain tensor $\varepsilon_{zz}$ of ZnSe nanorod with a flat ZnS shell. **j** The calculated elastic strain tensor $\varepsilon_{zz}$ of ZnSe nanorod with a helical ZnS shell. **k** Mapping of energy differences $\Delta E$ between the total surface and strain energy for different shell morphologies compared to the flat shell with the same shell volume corresponding to four monolayers. The different morphologies are represented by the number of islands along the perimeter (*m*) and the number of islands along the rod length (*k*). Several models with different *m* and *k* are illustrated to show the energy difference. The minimal energy is obtained for the helical-shell morphology

performed ZnS shell growth on CdSe/CdS seeded nanorods and on CdSe nanorods via the same method. Initial ZnS shell growth of ~2.6 monolayers on CdSe/CdS seeded nanorods produced uniform core/shell nanorods that remained smooth (Supplementary Fig. 13), followed by the appearance of three-dimensional ZnS islands as more ZnS was deposited (Fig. 2f and Supplementary Fig. 14). Similarly, ~1.4 monolayers of uniform ZnS were grown before the islands formation on the surface of CdSe nanorods (Fig. 2g and Supplementary Fig. 15).

For the epitaxial film growth on a flat substrate, the misfit strain energy at the interface increases as the film thickness[7,25]. For a specific film thickness, the stain energy is mainly determined by the lattice mismatch between the film and the substrate. Thereby, a thinner wetting layer is expected for a bigger lattice mismatch between a film and a substrate. The lattice mismatch between hexagonal ZnS and hexagonal ZnSe, CdS and CdSe is ~4.5%, 7.6%, and 11.2% respectively. To exclude the effect of core diameter on the critical thickness, ZnSe, CdS, and CdSe nanorods with similar diameters were used to extract the critical thickness of ZnS from corresponding TEM analysis (Supplementary Fig. 16–18). As displayed in Fig. 2h, the critical thickness decreases with the increase of the lattice mismatch between the core and the shell, clearly showing the critical thickness is inversely proportional to the lattice mismatch. This further establishes the unique SK growth on nanorods and offers principles for the design of such materials.

Three-dimensional island growth during the shell formation can be extended to different materials. Analogous islands growth of ZnSe was observed on CdSe/CdS seeded nanorods and ZnS nanorods (Supplementary Fig. 19 and 20). CdS growth on ZnSe-islands-decorated ZnS nanorods led to CdS growth around ZnSe islands, producing complex hetero-nanorods with larger islands (Supplementary Fig. 21). This further proves the formation of three-dimensional ZnSe islands on ZnS nanorods.

The shell growth of ZnS enhances the stability of ZnSe nanorods, making these nanorods suitable as photoinitiators for radical photo-polymerization of acrylamide in aqueous solution[26]. ZnSe nanorods did not show any photocatalytic activity under aerobic conditions (Fig. 2i), probably because of the surface oxidation of ZnSe nanorods. Indeed, ZnSe oxidation was observed in 24 h in aerobic conditions, manifested in the solution becoming reddish due to the formation of selenium oxyanions etched off from the rod[27]. Meanwhile, ZnSe/ZnS core/islands-shell nanorods maintained its stability and high photo-initiation capability in aerobic conditions. We did not observe any significant change in their absorption spectrum following exposure to oxygen.

**Strain-induced growth of helical shell**. We next discuss the intriguing ZnSe/ZnS core/helical-shell nanorods and their formation mechanism. High-resolution HAADF-STEM imaging is shown in Fig. 3a and using inverse fast Fourier transform (IFFT) filtered images (Fig. 3d), the magnified image of area e proves that the crystal structure of ZnSe nanorod remained hexagonal. Three magnified images selected at different locations along the helical shell manifest consistent atomic arrangement with the ZnSe core (f–h in Fig. 3), indicating the coherent nature of the shell growth. We also note that there is no apparent correlation of the helical pitch with stacking faults (Fig. 3a). Instead, we consider that the formation mechanism of the helical-shell growth is driven by strain. In MBE grown SK dots, the strain field was employed to explain the QDs formation. Moreover, the effects of existing islands in one QD layer was employed to generate coupled QDs in a vertical stack where the position of the underlying dots serves as the favorable nucleation site for the QDs in the next layer[28].

To unveil the transformation mechanism of the ZnS shell morphology on ZnSe nanorods from islands-shell to helical-shell, we applied a finite element simulation to resolve the strain inside

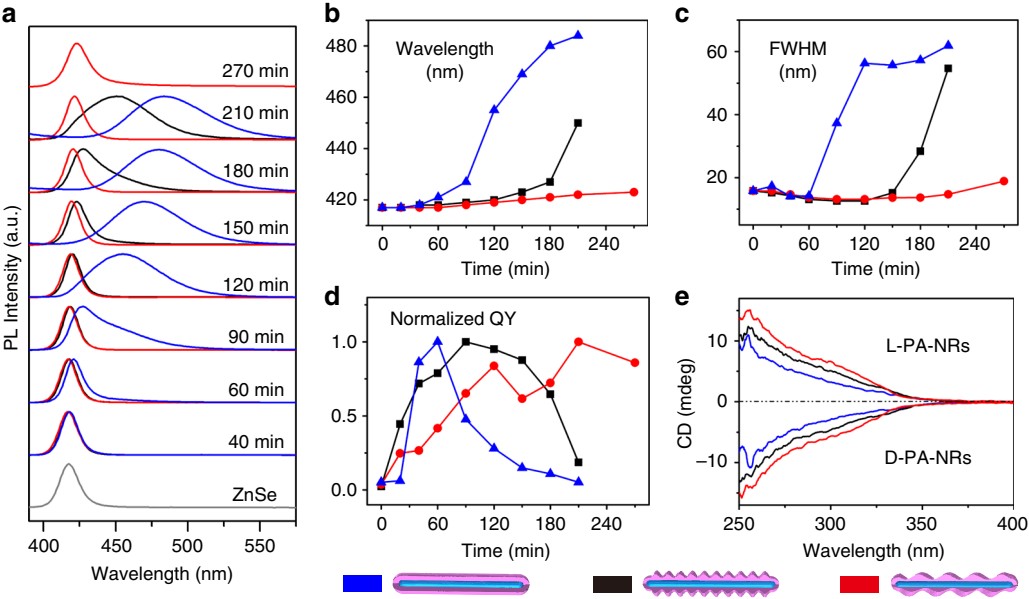

**Fig. 4** Optical properties of ZnSe/ZnS core/shell nanorods with different shell morphologies. Comparing spectra were obtained for similar growth conditions aside from precursor reactivity at different reaction times (blue—flat-shell, black—islands-shell, red—helical-shell). **a** Emission spectra. **b** Emission peak wavelength. **c** FWHM of emission spectra. **d** Normalized emission quantum yield. The normalization is performed based on the respective highest values. **e** CD spectra after chiral ligand exchange. The core/shell nanorods samples were transferred to water by using L- and D-penicillamine

the core/shell structure and also taking into account the surface energy (see Supplementary Methods and Supplementary Table 1 for details). In an epitaxial shell growth process, the shell attempts to adapt its lattice constant to that of the core. Figure 3i presents the elastic strain tensor $\varepsilon_{zz}$ in the case of a flat ZnS shell growth (four monolayers shell thickness) on ZnSe nanorods. Since ZnSe has higher lattice constant, the strain in the core is negative while in the shell it is positive. The growth of several monolayers of the shell accumulates the strain energy at the core-shell interface. In comparison, the strain is relaxed by forming a helical-shell structure as seen in Fig. 3j. In the thermodynamic limit, the actual structure is that with the minimal total energy, taking into account also the surface energy which is obviously minimized for the flat shell architecture.

It is the balance of the surface energy and the strain which dictates the actual stable morphology. The critical thickness for transforming from flat shell growth to other shell morphologies with higher surface area is at the point in which the relaxation of the strain energy exceeds the increased surface energy. The morphology which minimizes the sum of the strain energy and the surface energy will be favored. The critical point of this balance was calculated to take place at a thickness of 3.6 monolayers. Below this thickness, the total strain and surface energy per particle in the case of a flat shell was always smaller compared to any other oscillating morphology. However, when the shell exceeds 3.6 monolayers, the flat shell is no longer the lowest energy morphology. These results are correlated with the experimental results, showing that the critical wetting layer thickness is ~3.3 monolayers.

More intricate details of this balance between strain and surface energy contributions are garnered from Fig. 3k that presents the energy difference ($\Delta E$) between the total surface and strain energy per particle of different shell morphologies with the same shell volume compared to the flat shell corresponding to 4 monolayers. Different shell morphologies were defined using the general Equation (1):

$$(1)\quad r(\theta, z) = r_{flat} \cdot S + c \cdot \sin\left(m\theta + \frac{2\pi k}{L_{rod}} z\right)$$

where $r(\theta,z)$ is the distance of the outer shell surface from the rod axis, $r_{flat}$ is the shell radius in case of flat morphology, $S$ is a scaling factor which maintains the shell volume constant in the case of un-flat shell, $c$ is the oscillations amplitude of the perturbation in the shell morphology, $m$ is equivalent to the number of islands along the perimeter of the rod, $k$ is equivalent to the number of islands along the rod length and $L_{rod}$ is the rod length. In general, bigger $m$ and $k$ lead to more relaxation of strain energy but higher surface energy. The optimal morphology according to the simulation is obtained for $m = 1$ and $k = 4$ which represents a helical structure, similar to the experimental morphology seen in Fig. 3c. The formation of the helical shell necessitated the use of slow growth rate to reach the thermodynamicgrowth limit as corroborated by the simulations.

**The effect of shell morphology on optical properties**. A clear indication of the minimization of the total strain and surface energy for the thermodynamic favored morphologies can be found in comparison of the optical properties (Fig. 4 and Supplementary Fig. 22). The band gap of ZnS straddles that of ZnSe (Type-I band alignment). Bare ZnSe nanorods displayed a narrow emission at 421 nm with a low quantum yield (QY) of ~1.5%. For core/flat-shell nanorods, the band edge emission of ZnSe remained narrow after the ZnS deposition in the first 60 min, during which the QY increased to ~25%. Upon further ZnS growth, a broad emission emerged and became dominant and the QY rapidly decreased. These features are assigned to the formation of strain-induced defects at the core/shell interface formed during the fast deposition of the flat shell, consistent with the result of kinetically controlled growth.

For the core/islands-shell nanorods, the emission full width at half maximum (FWHM) remained narrow while the QY steadily increased up to maximal values of ~50% (at reaction times between 90 and 120 min). At this stage the ZnS shell reached the critical thickness upon which the islands growth just started. The emission polarization anisotropy was measured to be between 0.15 and 0.2 upon excitation at the short-wavelength range (Supplementary Fig. 23). These Cd-free, highly luminescent ZnSe/ZnS core/shell nanorods thus show potential as short-wavelength

emitters for linearly polarized emission[15]. Further growth of the ZnS shell led to gradual quenching of the emission intensity accompanied by an increasingly enhanced trap emission. Although the islands growth is beneficial to release the misfit strain energy, the actual thickness of the ZnS shell still increased during the islands growth, producing interfacial defects.

For core/helical-shell nanorods, initial uniform ZnS shell growth enhanced the emission QY to ~35% after 120 min when the shell reached the critical thickness. The ensuing growth of the helical shell maintained a narrow FWHM and stable QY levels even when the helical nanorods reached a comparable size to the samples with flat-shells and islands-shells morphology. This is in line with the helical shell being the energetically favorable morphology manifesting coherent growth, taking place under thermodynamic conditions.

Semiconductor nanocrystals with chirality attract increasing attention because of their potential intriguing properties such as circularly polarized luminescence[29,30]. However, the helical ZnSe/ZnS nanorods did not show optical activity, indicating that they are racemic. This is not surprising since there is no intrinsically preferred handedness for the helical ZnS shell. Both handedness directions can be identified on different nanorods (Supplementary Fig. 8). However, some indication of the unique chiral properties of the helical-shell rods could be observed by chiral ligand exchange[31]. Hydrophilic chiral ligands were introduced enabling phase transfer to water of these three samples with different morphologies. They all show some circular dichroism (CD) optical activity, resulting from the interaction of the surface zinc atoms with the chiral ligands[32] (CD spectra of D- and L-penicillamine aqueous solution were shown in Supplementary Fig. 24). ZnSe/ZnS nanorods with the helical-shell morphology manifest the highest CD values among the three morphologies (Fig. 4e).

Our study demonstrates a strain-controlled transformation of shell morphology from flat-, to islands- (colloidal SK growth) and eventually to helical-shell on one-dimensional nanorods, simply by changing the shell growth rates via tuning the precursors' reactivity. The helical morphology is energetically favored because of the relaxation of the strain energy during the coherent shell growth, approaching to a thermodynamic growth regime. We envision that the shell adjustment can be expanded to other semiconductor nanoparticles, where employment of thermodynamic growth conditions can be used for avoiding interfacial defects which are detrimental to the optical properties with clear applicative importance as well.

## Methods

**Chemicals**. Diethylzinc solution (1.0 M in hexane), zinc acetate (anhydrous, 99.99%), zinc oxide (ZnO, 99.0%), 1-octadecene (ODE, 90%), oleic acid (OA, 90%), selenium (99.99%), 1-octanethiol (≥98.5%), oleylamine (OLA, 70%), cadmium oxide (CdO, ≥99.99%), trioctylphosphine oxide (TOPO, 99%), diphenylphosphine (DPP, 98%), tetrakis(acetonitrile)copper(I) hexafluorophosphate (97%) and zinc chloride (dry, ≥98.5%) were purchased from Sigma. Trioctylphosphine (TOP, 97%) was purchased from Strem. Octadecylphosphonic acid (ODPA, >99%), tetradecylphosphonic acid (TDPA, >99%) and hexylphosphonic acid (HPA, >99%) were purchased from PCI synthesis. All chemicals were used as received without any further purification.

**Preparation of precursors**. 0.1 M zinc oleate solution was prepared by heating 10 mmol of zinc oxide in 20 mL of oleic acid (the mole ratio between Zn and oleic acid is 1:6.3) at 300 °C under Ar for 1 h. Then the flask was cooled down to 80 °C, in which 20 mL of oleylamine and 60 mL of ODE were added. The solution was then degassed under vacuum at 100 °C for 1 h. Zinc oleate solution (0.1 M) with controlled slow reactivity was prepared by changing the mole ration between Zn and oleic acid from 1:6.3 to 1:10 (or 12). The volume change was compensated by ODE. For the preparation of zinc oleate solutions (0.1 M) with controlled high reactivity, 917.5 mg of zinc acetate (5 mmol), 6.31 mL of oleic acid (20 mmol) and 28.7 mL of ODE were first degassed at 120 °C for 30 min to get a clear solution, followed by heating the solution to 250 °C for 1 h under Ar. Then the flask was cooled down to

80 °C, in which 15 mL of oleylamine were added. The solution was then degassed under vacuum at 100 °C for 1 h. 0.5 M cadmium oleate solution was synthesized by heating 10 mmol of cadmium oxide in 20 mL of oleic acid (the mole ratio between Cd and oleic acid is 1:6.3) at 160 °C under Ar until a colorless solution was obtained. The solution was then degassed under vacuum at 100 °C for 1 h.

**Synthesis of ZnSe nanorods**. First $Zn_4$ clusters ($(Me_4N)_2[Zn_4(SPh)_{10}]$) were prepared via a reported procedure and dried under vacuum[13,33]. 12.0 mL of OLA and 20.0 mL of ODE were degassed under vacuum at 110 °C in a three-neck flask for 1 h and cooled down to room temperature under Ar. Then 4.0 mL of diethylzinc solution (1.0 M), 120 mg of $Zn_4$ clusters (dissolved in 4.0 mL of DMF) and 6.0 mL of DPP-Se (0.67 M, in toluene) were injected successively into the three-neck flask. The solution was heated slowly to 65 °C and kept at 65 °C for 48 h to grow ZnSe magic-sized clusters (MSCs). The growth of ZnSe MSCs was manifested by the well-defined peak at 312 nm in the absorption spectrum. Then the solution was heated to 240 °C in a rate of 6 °C/min and stayed at 240 °C for ~2 h to obtain ZnSe nanorods through oriented attachment. The diameter of ZnSe nanorods was further increased by heating the solution to 280 °C, during which 50.0 mL of ODE were injected when the solution started to become turbid. After the addition of ODE, the solution was further heated to 310 °C to increase the diameter of ZnSe nanorods. The synthesis was quenched when the desired diameter was obtained. The length of ZnSe nanorods was tuned by the amount of $Zn_4$ clusters. Decreasing the amount of $Zn_4$ clusters will produce longer ZnSe nanorods. For example, ZnSe nanorods with the length of ~60 nm were obtained when adding 40 mg of $Zn_4$ clusters. ZnSe nanorods were washed by hexane and acetone and redispersed in hexane. It should be noted that the above washing process was performed in an inert atmosphere in a glovebox filled with nitrogen. The purified ZnSe nanorods solution was stored in a glovebox.

**Synthesis of ZnSe/ZnS core/shell nanorods**. The molar absorptivity at 350 nm was measured and used to calculate the concentration of ZnSe nanorods[34]. 1.5 mL of ODE, 1.5 mL of oleic acid, 1.5 mL of oleylamine and ~10 nmol of ZnSe nanorods (in hexane) were loaded in a four-neck flask. The flask was degassed under vacuum at 90 °C for 1 h and then heated to 310 °C under Ar. The sulfur precursor solution was prepared by diluting 0.338 mL of 1-octanethiol (1.3 equivalent amounts of zinc) in 14.7 mL of ODE. When the temperature reached 240 °C, zinc and sulfur precursor solutions were injected in the flask at a rate of 3 mL/h separately through a syringe pump. The growth solution was further annealed at 310 °C for 5 min when the precursors with desired amounts were injected. The shell morphology including flat-, islands- and helical-shell was tuned by injecting zinc oleate with the molar ratio between zinc/oleate acid of 1/4, 1/6.3 and 1/10, respectively. ZnSe/ZnS nanorods were washed with hexane and acetone and redispersed in hexane. It should be noted that all the manipulations above were performed in an inert atmosphere in a glovebox filled with nitrogen or on a Schlenk line. The purified ZnSe/ZnS nanorods solution was stored in a glovebox.

**Shell growth of ZnS on CdSe/CdS seeded nanorods and CdSe nanorods**. CdSe/CdS seeded nanorods and CdSe nanorods were first synthesized according to literature methods[35–37]. The shell growth of ZnS on CdSe/CdS seeded nanorods and CdSe nanorods were similar to the synthesis of ZnSe/ZnS core/shell nanorods.

**Synthesis of ZnS nanorods via a cation exchange reaction**. CdS nanorods were first synthesized according to a literature protocol[36,38]. In the glove box, purified CdS nanorods (~1.43 mmol of Cd) were dispersed in 250 mL of toluene. A solution prepared by dissolving 3.2 g of tetrakis(acetonitrile)copper(I) hexafluorophosphate in 110 mL of methanol (the mole ratio between Cu and Cd is 6:1) was added in the CdS nanorods solution under stirring. The obtained $Cu_2S$ nanorods were washed with methanol and toluene for 3 times and redispersed in 60 mL of TOP. For the cation exchange from Cu to Zn, 2.34 g of $ZnCl_2$ (17 mmol), 108 mL of oleylamine and 180 mL of ODE were loaded in a three-neck flask. The mole ratio between Zn and Cu is 6:1. The solution was degassed under vacuum at 90 °C for 30 min. Under Ar, the solution was heated to 150 °C, followed by the injection of $Cu_2S$ nanorods solution (in TOP). The heating was kept for 10 min to complete the cation exchange reaction.

**ZnSe shell growth on CdSe/CdS seeded nanorods**. 184 mg of zinc acetate (1 mmol), 1.13 g of oleic acid (4 mmol) and 6.0 mL of ODE were added in a three-neck flask. The solution was first degassed under vacuum at 120 °C for 30 min and then heated to 250 °C for 1 h under Ar for the formation of zinc oleate. After that, the solution was cooled down to 60 °C, followed by the introduction of CdSe/CdS nanorods (in chloroform). Then the flask was degassed again to get rid of the chloroform at 60 °C for 30 min and heated to 300 °C. When the temperature reached 160 °C, 48 mg of Se (0.6 mmol) dissolved in 2.0 mL of TOP were injected in the flask at a rate of 6 mL/hr. After the injection, the solution was kept at 300 °C for 2 min and cooled down to room temperature. The similar protocol was used for the shell growth of ZnSe on ZnS nanorods.

**Shell growth of CdS on CdSe, ZnS, ZnSe, and ZnS/ZnSe nanorods**. Ten nanomols of CdSe, ZnS, ZnSe or CdS/ZnSe nanorods were added in a mixture of 1.5 mL of ODE, 1.5 mL of oleic acid and 1.5 mL of oleylamine in a three-neck flask. After degassing under vacuum to remove the low boiling point solvent, the solution was heated to 310 °C under Ar. When the temperature reached 240 °C, a desired amount of Cd precursor (Cd oleate) and sulfur precursor (1-octanethiol, 1.2 equivalent amounts of cadmium) were injected separately. The growth solution was further annealed at 310 °C for 10 min after the injection.

**Photo-polymerization measurements**. Cleaned ZnSe nanorods and ZnSe/ZnS core/islands-shell nanorods (~1 nmol, in 1.0 mL of chloroform) were first transferred to water through polymer coating with polyethylenimine (PEI) (0.15 g; MW 25,000 g/mol) in chloroform (1.0 mL) for 1 h[26,39]. Cyclohexane was added to precipitate the particles. Then, the particles are washed from excess of PEI and from the original organic ligands with chloroform/cyclohexane. The nanorods were finally redispersed in TDW. We studied polymerization kinetics of acrylamide in aqueous solution to determine the efficiency of ZnSe and ZnSe/ZnS nanorods as photo-initiators. Hundred microlitres of monomer solution (10.0 g of acrylamide and 1.0 g of ethoxylated trimethylolpropane triacrylate were dissolved in 5 mL of TDW) was mixed with 100 μL of nanorods aqueous solutions with the help of vortex in dark. Then 30 μL of this solution placed on the attenuated total reflection (ATR) crystal plate of a FTIR spectrophotometer (Nicolet iS50 FTIR Spectrometer). The sample drop was irradiated with 405 nm from Monochromatic UV LED (MEMCOM, Jerusalem, Israel, intensity of 30 mW/cm$^2$) while infrared spectra were recorded after every 10 s, for a total duration of 60 sec. The polymerization degree was abstracted from the decrease of the Fourier transform infrared (FTIR) absorption peaks of methylene group vibrations at 988 cm$^{-1}$ (the out-of-plane bending mode of the =C–H unit) normalized to the C=O stretch peak at 1654 cm$^{-1}$ as an internal reference.

**Characterizations**. UV–vis absorption and emission spectra were recorded on a JASCO V-570 spectrometer and Varian Cary Eclipse spectrophotometer respectively. The sample fluorescence quantum yields were evaluated in comparison with coumarin 1 in ethanol as the reference (excited at 360 nm). Photo-selection excitation measurements were performed on Edinburgh Instruments FLS920 Fluorometer. XRD measurements were performed on a Phillips PW1830/40 diffractometer using the Cu Kα photons. TEM and HRTEM images were obtained on FEI Tecnai G2 Spirit Twin T-12 TEM and F20 G2 HRTEM. High-resolution HAADF STEM images and elemental mapping were obtained on FEI Themis G3 at 300 keV.

## Data availability
The data that support the findings of this study are available from the corresponding author upon request.

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

## Acknowledgements
The research leading to these results has received financial support from the Israel Science Foundation (ISF, Grant No. 1867/17) in the frame of the Alternative Fuels Program. We thank Lior Verbitsky for the help on data analysis. U.B. thanks the Alfred & Erica Larisch memorial chair. Y.E.P. acknowledges support by the Ministry of Science and Technology & the National Foundation for Applied and Engineering Sciences, Israel.

## Author contributions
B.J. and U.B. designed the experiments. B.J. performed the synthesis and materials characterization. N.W. and B.J. carried out photo-polymerization experiments. Y.E.P.

performed the calculation. S.R. and I.P. assisted with HR-STEM measurements and structural analysis. B.J., Y.E.P., and U.B. wrote the manuscript with help from the other authors.

## Additional information

**Competing interests:** The authors declare no competing interests.

