## [Peer Review File · Nature Communications]

Parts of this peer review file have been redacted as indicated to maintain the confidentiality of unpublished data.

Reviewers' comments:

Reviewer #1 (Remarks to the Author):

The paper by Banin and co-workers solves a long lasting problem regarding the shell growth on elongated semiconductor nanostructures. While the invention of spherical core-shell nanocrystals was the key towards the breakthrough of nanocrystals for several applications (e.g. in solid-state lighting), the shell growth on 1D nanorods and nanowires was much less successful in recent years. Several research groups throughout the world showed attempts to grow such shells, but all of them ended up with "odd-structures", i.e. islands, needles of more or less irregularly shaped shell structures instead of close shells on nanowires. Some of these attempts indeed led to benefits in the optical performance of the 1D systems, such as increase in QY or long-term photo stability. However, the reason for the different shell morphologies or even a strategy to direct the growth mode in a certain direction in order to come up with a desired morphology has not been achieved so far.

The current contribution is based on the idea, that the balance between interfacial strain and surface energy governs the shell growth mode on elongated semiconductor nanostructures, prepared by wet chemistry. While this idea is not necessary new in the first place, the authors show a on a whole bunch of examples that it needs a careful adjustment of reaction parameters to realize very different shell morphologies by using this fundamental concept. So the quality of the samples is excellent throughout the paper: the authors start with almost perfectly homogeneous 1D ZnSe-, CdSe- and CdS-rods, then they carefully adjust the reaction parameters to enter the different growth regimes (kinetic/thermodynamic) for shell growth, and finally they show highest quality TEM- and XRD-results for sample characterization. Most importantly, they use a sophisticated theory to calculate the influences of different material parameters on the morphology and find very good agreements with the observed results. This finally led a relative simple formula (eq.: 1) which will certainly be used by other groups to adjust the material parameters in order to prepare different 1D-nanostructures with distinct morphologies.

The most prominent result is certainly the realization of chiral shells around 1D nanostructures. So far, chirality on 1D nanostructures has only been introduced by molecular surface modification using chiral ligand. The helical shell structures shown here are very defined and homogeneous, which will certainly open up a new field in nanoscale research.

The whole paper is very well written and the details are accurately described. So there is not a sign of "hidden secrets". Hence, the results should be easily reproducible by other groups. Especially the supporting information is very informative. However, somehow there is even too much information, which is kind of overloading the paper. For example, details on growth modes on different regions of the rods or information on the photo catalytic properties could be part of follow-up papers. Instead,

the part dealing with the evolving optical properties is compressed only on the last 3 pages of the paper. Also, here they show only results on the ZnSe/ZnS-system, while in the first part of the paper they emphasize that the method is universal and demonstrate the generality for different systems. For example, it would be interesting to see a comparison also for the CdSe/CdS-DRs with and without a (chiral) ZnSe shell (cf. S19), where the PL- data could even be taken from individual structures.

In summary, the paper is highly interesting for specialists in structure modification in nanoscale research, but also to a broader readership of NATURE-communications, since it translates basic concept of strain-induced growth initially developed for planar structures to highly curved 1D-nanomaterials. They demonstrate on several systems the generality of their approach, and show how very unique structures such 1D nanostructures with a chiral shell can be achieved. If the first part would be written more tightly and the generality could be shown also for the optical properties, evt. in addition with exemplary data for single nanorod PL, the paper should certainly be published in NATURE communications.

Reviewer #2 (Remarks to the Author):

This manuscript from Banin and coworkers details the formation of ZnSe-ZnS core-shell nanorods with unique shell morphologies. The hypothesis is that control of ZnS precursor reactivity towards the thermodynamic limit enables the control of shell morphology by minimizing strain and surface energy. Ultimately a helical shell morphology is obtained and characterized. I think that the observation of helicity without a chiral template is intriguing and of general interest to the scientific community, however I have some reservations about the primary conclusions and novelty of the observed shell growth. Several specific points for consideration are detailed below.

1. The shell morphologies shown in Figure 1 are quite remarkable and I recalled seeing structures like that in a previous paper - specifically, a 2016 JACS paper titled "Metal Oleate Induced Etching and Growth of Semiconductor Nanocrystals, Nanorods, and Their Heterostructures". In this paper the authors demonstrate the formation of very similar structures starting with CdS nanorods. Addition of Zn oleate is shown to etch CdS nanorods anisotropically and island formation on the nanocrystals is observed upon further addition of precursors. This seems like an incredibly relevant piece of work that may shed light on some of the microscopic chemical transformations taking place during this shell growth observed here. The authors cite this generically in the introduction, but do not draw any relationship between the structures grown here and the structures observed in that paper. I believe this would be necessary for a balanced presentation of the current work.

2. Reference 13 is not yet published and so additional details are necessary with respect to the oriented attachment synthesis of the ZnSe nanorods.

3. The authors state that they adjust the reactivity of zinc oleate by "changing the molar ratio between zinc and oleic acid" and cite reference 18. I think for a general audience the important point here is that the reactivity is being modulated by changing the solubility of the zinc precursor, thus limiting zinc availability. This should be stated.

4. The authors state at the top of page 4 that their first 1/4 molar ratio condition is "the result of kinetically controlled shell growth". I don't think this is an obvious conclusion based on the data that has been presented to this point. This should be formulated as a hypothesis or further justified in some way.

5. The discussion that begins at line 91 is quite convoluted. In that paragraph the authors state, "The ZnS shell maintained uniformity upon further addition of shell precursors." and then "Further ZnS shell growth produced nanorods with the appearance of small bulges...". More quantitative metrics are needed for clarity here. "Further" doesn't tell the reader anything useful.

Response to referees for Manuscript Nature Communications NCOMMS-18-26550-T
Entitled:

Strain-controlled shell morphology on quantum rods: from flat, to islands, and to helical shell architectures

We would like to thank the two referees for their constructive comments. Please find below a point by point response to the referees' comments.

Referee #1 (Remarks to the Author):

The paper by Banin and co-workers solves a long lasting problem regarding the shell growth on elongated semiconductor nanostructures. While the invention of spherical core-shell nanocrystals was the key towards the breakthrough of nanocrystals for several applications (e.g. in solid-state lighting), the shell growth on 1D nanorods and nanowires was much less successful in recent years. Several research groups throughout the world showed attempts to grow such shells, but all of them ended up with “odd-structures”, i.e. islands, needles of more or less irregularly shaped shell structures instead of close shells on nanowires. Some of these attempts indeed led to benefits in the optical performance of the 1D systems, such as increase in QY or long-term photo stability. However, the reason for the different shell morphologies or even a strategy to direct the growth mode in a certain direction in order to come up with a desired morphology has not been achieved so far.

The current contribution is based on the idea, that the balance between interfacial strain and surface energy governs the shell growth mode on elongated semiconductor nanostructures, prepared by wet chemistry. While this idea is not necessary new in the first place, the authors show a on a whole bunch of examples that it needs a careful adjustment of reaction parameters to realize very different shell morphologies by using this fundamental concept. So the quality of the samples is excellent throughout the paper: the authors start with almost perfectly homogeneous 1D ZnSe-, CdSe- and CdS-rods, then they carefully adjust the reaction parameters to enter the different growth regimes (kinetic / thermodynamic) for shell growth, and finally they show highest quality TEM- and XRD-results for sample characterization. Most importantly, they use a sophisticated theory to calculate the influences of different material parameters on the morphology and find very good agreements with the observed results. This finally led a relative simple formula (eq.: 1) which will

certainly be used by other groups to adjust the material parameters in order to prepare different 1D-nanostructures with distinct morphologies.

The most prominent result is certainly the realization of chiral shells around 1D nanostructures. So far, chirality on 1D nanostructures has only been introduced by molecular surface modification using chiral ligand. The helical shell structures shown here are very defined and homogeneous, which will certainly open up a new field in nanoscale research.

The whole paper is very well written and the details are accurately described. So there is not a sign of "hidden secrets". Hence, the results should be easily reproducible by other groups. Especially the supporting information is very informative. However, somehow there is even too much information, which is kind of overloading the paper. For example, details on growth modes on different regions of the rods or information on the photo catalytic properties could be part of follow-up papers. Instead, the part dealing with the evolving optical properties is compressed only on the last 3 pages of the paper. Also, here they show only results on the ZnSe/ZnS-system, while in the first part of the paper they emphasize that the method is universal and demonstrate the generality for different systems. For example, it would be interesting to see a comparison also for the CdSe/CdS-DRs with and without a (chiral) ZnSe shell (cf. S19), where the PL-data could even be taken from individual structures.

In summary, the paper is highly interesting for specialists in structure modification in nanoscale research, but also to a broader readership of NATURE communications, since it translates basic concept of strain-induced growth initially developed for planar structures to highly curved 1D-nanomaterials. They demonstrate on several systems the generality of their approach, and show how very unique structures such 1D nanostructures with a chiral shell can be achieved. If the first part would be written more tightly and the generality could be shown also for the optical properties, evt. in addition with exemplary data for single nanorod PL, the paper should certainly be published in NATURE communications.

Reply: We thank the referee for the highly positive recognition of the innovative part of our work that "solves a long lasting problem regarding the shell growth on elongated semiconductor nanostructures". Indeed, the paper lays out a complete story on the challenge and the unique outcomes of such shell growth on nanorods, which is

needed to convey clearly the story, to show its generality, and to demonstrate the structure-function consequences of the shell growth motif on the optical properties of the core/shell nanorods. Therefore – we respectfully did not shorten the manuscript as this would compromise clarity.

With regards to the functionality of the core/shell nanorods as a heavy-metal-free photo-initiator, it is important in our opinion to also demonstrate the benefit of the shell on yielding stable photo-initiator function. In the context of demonstrating potential use of such architectures in emerging applications, this demonstration offers a nice example.

With regards to the suggestion of the referee to demonstrate the generality of the shell growth modes and their effects, we demonstrated in the first part the generality of the islands-shell growth behavior of ZnS on colloidal ZnSe, CdSe/CdS seeded nanorods and CdSe nanorods *via* the same method.

The revelation of the importance of reaching the thermodynamic limit of shell growth, and its unique consequences does indeed have additional far-reaching implications, which will lead to future studies. Specifically, the referee suggested investigating more deeply the optical properties of CdSe/CdS dot-in-rods (DRs) with and without a (chiral) ZnS shell. However, the CdSe/CdS DRs are already a core/shell system, and the effect of additional ZnS shell morphology on optical properties of CdSe/CdS DRs might be minor. We believe that such a study – while potentially interesting, is beyond the scope of the present contribution.

[REDACTED]

Referee #2 (Remarks to the Author):

Comments: *This manuscript from Banin and coworkers details the formation of ZnSe-ZnS core-shell nanorods with unique shell morphologies. The hypothesis is that control of ZnS precursor reactivity towards the thermodynamic limit enables the control of shell morphology by minimizing strain and surface energy. Ultimately a helical shell morphology is obtained and characterized. I think that the observation of helicity without a chiral template is intriguing and of general interest to the scientific community, however I have some reservations about the primary conclusions and novelty of the observed shell growth. Several specific points for consideration are detailed below.*

Reply: We thank the referee for the positive assessment of the manuscript acknowledging the intriguing aspects of the work. We have addressed all the points raised by the referee and modified the paper accordingly as addressed below in the detailed point-by-point response to the comments.

1. *The shell morphologies shown in Figure 1 are quite remarkable and I recalled seeing structures like that in a previous paper - specifically, a 2016 JACS paper titled "Metal Oleate Induced Etching and Growth of Semiconductor Nanocrystals, Nanorods, and Their Heterostructures". In this paper the authors demonstrate the formation of very similar structures starting with CdS nanorods. Addition of Zn oleate is shown to etch CdS nanorods anisotropically and island formation on the nanocrystals is observed upon further addition of precursors. This seems like an incredibly relevant piece of work that may shed light on some of the microscopic chemical transformations taking place during this shell growth observed here. The authors cite this generically in the introduction, but do not draw any relationship between the structures grown here and the structures observed in that paper. I believe this would be necessary for a balanced presentation of the current work.*

Reply: It is a good suggestion to compare the two works. Indeed, at first sight the morphology of islands-shell in current work ensembles the islands-decorated structures observed in the paper the referee mentioned. In that paper, the authors carefully studied the unexpected etching of semiconductor nanorods induced by metal oleate. Specifically, CdS nanorods were anisotropically etched by zinc oleate. The reaction for Zn oleate induced etching of CdS NRs can be described below (Ref 1):

$(\text{CdS})_{\text{ligand}} + \text{Zn}^{2+}(\text{OI}^-)_2 \rightarrow \text{Cd}^{2+}(\text{OI}^-)_2 + \text{S}^{n-} + (\text{Zn}^{2+})_{\text{ligand}} + (\text{Zn}^{2+})_{\text{adsorbed}}$ The followed addition of Se at high temperature led to the growth of CdSe QDs on the

tips and sides of CdS nanorod. When excess of oleic acid was added, alloyed CdZnSe nanoparticles were grown on the tips and sides of CdS nanorods. However, the authors did not focus on the formation mechanism of the CdSe and CdZnSe nanoparticles growth as islands instead of a shell.

In the current work, ZnS shell was grown on ZnSe nanorods by injecting Zn precursor (zinc oleate) and S precursor (1-octanethiol). The shell precursors were added slowly at high temperature and then reacted for the shell growth, thus the actual concentrations of both shell precursors in the solution were low. Additionally, the ZnSe nanorods studied here possess the same cation ions as the zinc precursor and therefore the addition of zinc oleate will suppress their etching. Therefore etching is not a significant process during the synthesis of ZnSe/ZnS core/islands-shell.

Thanks to the well-controlled shell growth on elongated nanostructures, we systematically studied the evolution of shell morphology during the synthesis of ZnSe/ZnS core/shell nanorods. As shown in the manuscript, the shell morphology can be transformed from flat, to islands and to helical shell by reducing the shell growth rate. In the case of islands-shell growth, we demonstrated the growth of a *wetting layer* before the islands emerged, a distinguishing feature of colloidal Stranski-Krastanov (SK) growth mode. We also show the critical thickness decreases with the increase of the lattice mismatch between the core and the shell, indicating the strain between the core and the shell plays a critical role for the islands-shell growth.

To address this comment - in the modified manuscript, we now added discussion of the paper mentioned by the referee to achieve a balanced presentation of the current work, as suggested.

Changes made: Added on page 8, at the end of 1st paragraph. While considering the location of the islands, we performed HRTEM analyses which excluded the possibility that the formation of islands-shell is related to the stacking faults in the core rod (Fig. S10 and S11). It should be noted that Oh *et al.* reported on islands-decorated nanostructure by growing CdSe quantum dots (QDs) on CdS nanorods anisotropically etched by zinc oleate¹². However, in the present work the shell precursors were added slowly at high temperature and then reacted for the shell growth, thus the actual concentrations of both shell precursors in the solution were low. Additionally, the ZnSe nanorods studied here possess the same cation ions as the zinc precursor and therefore the addition of zinc oleate will suppress their etching. Therefore etching is not a significant process during the synthesis of ZnSe/ZnS core/islands-shell.

2. Reference 13 is not yet published and so additional details are necessary with respect to the oriented attachment synthesis of the ZnSe nanorods.

Reply: The paper of the synthesis of ZnSe nanorods has been published in the meantime in JACS (Ref 2). Title: Controlling anisotropic growth of colloidal ZnSe nanostructures (DOI: 10.1021/jacs.8b05941). It is now properly and fully cited.

The synthesis scale here was 4-time bigger than that in the new published paper. Actually, the protocol was improved by introducing more 1-octadecene (ODE) during the synthesis to obtain a better colloidal dispersion of ZnSe nanorods in the current manuscript (see the experimental section). We also added more details about the relation between the amount of Zn₄ clusters and the length of obtained ZnSe nanorods. We believe the synthesis of ZnSe nanorods will be reproducible based on the protocol presented in this work.

Changes made: Added on page 19, 2nd paragraph. The length of ZnSe nanorods was tuned by the amount of Zn₄ clusters. Decreasing the amount of Zn₄ clusters will produce longer ZnSe nanorods. For example, ZnSe nanorods with the length of ~60 nm were obtained when adding 40 mg of Zn₄ clusters. ZnSe nanorods were washed by hexane and acetone and redispersed in hexane.

Added on page 24. Ref 13. J. Ning, J. Liu, Y. Levi-Kalisman, A. I. Frenkel, U. Banin, Controlling anisotropic growth of colloidal ZnSe nanostructures. J. Am. Chem. Soc. DOI: 10.1021/jacs.8b05941 (2018).

3. *The authors state that they adjust the reactivity of zinc oleate by "changing the molar ratio between zinc and oleic acid" and cite reference 18. I think for a general audience the important point here is that the reactivity is being modulated by changing the solubility of the zinc precursor, thus limiting zinc availability. This should be stated.*

Reply: The statement is added, as suggested.

Changes made: Added on page 3, 3rd paragraph. The reactivity of zinc oleate was adjusted in different experiments by changing the molar ratio between zinc and oleic acid as a powerful means to manipulate the shell growth morphology¹⁸. Larger excess of oleic acid can increase the solubility of the zinc precursor and thus limit zinc availability, producing zinc precursor with lower reactivity.

4. *The authors state at the top of page 4 that their first 1/4 molar ratio condition is "the result of kinetically controlled shell growth". I don't think this is an obvious conclusion based on the data that has been presented to this point. This should be formulated as a hypothesis or further justified in some way.*

Reply: We agree with the referee that "the result of kinetically controlled shell growth" is not an obvious conclusion based on the data that has been presented to this point. In the revised manuscript, we formulated this as a hypothesis at this point and further justified in the following discussion part.

In the discussion part about the formation mechanism of islands-shell growth (on Page 8 & 9), we performed control experiments, CdS shell growth on CdSe, ZnS and ZnSe nanorods using cadmium oleate. Relatively flat CdS shells were obtained in all

cases (Fig. S12) even when the CdS shell was very thick. The reactivity of cadmium oleate is higher than zinc oleate (Ref 1). The higher reactivity of cadmium oleate leads to the fast deposition of CdS shell, overcoming the dislocation nucleation by the formation of islands. These results indicate the flat-shell formation is a result of kinetically controlled shell growth. Besides, the QY of ZnSe/ZnS core/flat-shell nanorods first increased and then rapidly decreased as the shell thickness increased due to the formation of strain induced defects, which is also consistent with the result of kinetically controlled growth.

Changes made: Added on page 4, 1st paragraph. A substantially flat ZnS shell morphology is observed by transmission electron microscopy (TEM) (Fig. 1b, termed as core/flat-shell nanorods for simplification). Hypothetically, this is a result of the kinetically controlled shell growth process in these conditions, which will be further justified.

Added on page 8, 1st paragraph. The islands-shell growth of ZnS resembles the characteristics of the SK growth but on the finite rod surface. In this process, slowing the ZnS growth rate plays a key role in allowing dislocation nucleation of islands to release the interfacial strain energy unlike fast growth that results in the flat shell morphology (Fig. 1B). The mechanism of this type of growth reactivity control on shell growth is further substantiated by studying CdS shell growth on CdSe, ZnS and ZnSe nanorods using cadmium oleate with a 1/6.3 molar ratio between cadmium and oleic acid (the same molar ratio as in the case of ZnS islands-shell growth). Relatively flat CdS shells were obtained in all cases (Fig. S12) even when the CdS shell was very thick. The higher reactivity of cadmium oleate than zinc oleate¹² leads to the fast deposition of CdS shell, overcoming the dislocation nucleation by the formation of islands, indicating the flat-shell formation is a result of kinetically controlled shell growth.

Added on page 15, 1st paragraph. These features are assigned to the formation of strain induced defects at the core/shell interface formed during the fast deposition of the flat shell, consistent with the result of kinetically controlled growth.

5. *The discussion that begins at line 91 is quite convoluted. In that paragraph the authors state, "The ZnS shell maintained uniformity upon further addition of shell precursors." and then "Further ZnS shell growth produced nanorods with the appearance of small bulges...". More quantitative metrics are needed for clarity here. "Further" doesn't tell the reader anything useful.*

Reply: We thank the referee for pointing out this unclear description. Now we present the process in a more quantitative way.

Changes made: on page 4, 2nd paragraph. By reducing the reactivity (using zinc oleate with a 1/6.3 molar ratio between zinc and oleic acid)¹⁸, ZnSe nanorods with a thin ZnS shell are obtained initially in the first 60 min (Fig. 1c). The diameter of the nanorods increased by 1.1 nm to ~5.1 nm (Fig. S3a and S4). HRTEM image shows

that the shell grew in an epitaxial way. The ZnS shell maintained uniformity upon further addition of shell precursors (reaction time, 90 min). The diameter of the nanorods increased to 6.2 nm, corresponding to a shell thickness of ~3.5 monolayers of ZnS (the thickness of one monolayer of wurtzite ZnS along the [100] growth direction is 0.31 nm) (Fig. S3b). Further ZnS shell growth produced nanorods with the appearance of small bulges on the surface, indicating the inhomogeneity of the shell thickness (Fig. S3c, reaction time, 120 min). Adding more shell precursors led to thicker nanorods with increased roughness of the nanorod surface (Fig. S3d and e, reaction times between 150 and 180 min). When the amount added was equivalent to the flat shell case (Fig 1b) discussed above, a zigzag ZnS shell structure was obtained (Fig. 1d and S3f, reaction time, 210 min).

References used in this discussion:

- (1) Oh, N. & Shim, M. Metal oleate induced etching and growth of semiconductor nanocrystals, nanorods, and their heterostructures. *J. Am. Chem. Soc.* **138**, 10444-10451 (2016).
- (2) J. Ning, J. Liu, Y. Levi-Kalisman, A. I. Frenkel, U. Banin, Controlling anisotropic growth of colloidal ZnSe nanostructures. *J. Am. Chem. Soc.* DOI: 10.1021/jacs.8b05941 (2018).

REVIEWERS' COMMENTS:

Reviewer #1 (Remarks to the Author):

Dear Editors,

the authors wrote an extended response. So basically, I had just a few suggestions: shortening the first part, showing some PL on individual core shell rods and a demonstration for the generality of the approach.

- I can follow their arguments why they didn't shorten the first part.

- [redacted]

- But they didn't even mention the possibility of single dot spectroscopy, even though this seems to be a standard technique in the Banin lab. However, in reading the paper again I have to admit that this would also be beyond the scope of the paper, because it needs a lot of statistics and usually comes with a lot of new questions.

So in summary I am satisfied with the paper now - also with the way they dealt with detailed questions raised by my colleague referee.

In summary I guess the paper can be published in NATURE communications as it is right now.

Reviewer #2 (Remarks to the Author):

The authors have provided a detailed and thorough revision to their work in response to the reviewer comments. All of my comments have been adequately addressed and I believe the work is now suitable for publication in Nature Communications.